# Quartz Tuning Fork Resonance Tracking and application in Quartz Enhanced Photoacoustics Spectroscopy

**DOI:** 10.3390/s19245565

**Published:** 2019-12-16

**Authors:** Roman Rousseau, Nicolas Maurin, Wioletta Trzpil, Michael Bahriz, Aurore Vicet

**Affiliations:** IES, University Montpellier, CNRS, 34095 Montpellier, France; roman.rousseau@ies.univ-montp2.fr (R.R.); nicolas.maurin@ies.univ-montp2.fr (N.M.); wioletta.trzpil@ies.univ-montp2.fr (W.T.); michael.bahriz@umontpellier.fr (M.B.)

**Keywords:** quartz tuning fork, gas sensor, photoacoustic, beat frequency

## Abstract

The quartz tuning fork (QTF) is a piezoelectric transducer with a high quality factor that was successfully employed in sensitive applications such as atomic force microscopy or Quartz-Enhanced Photo-Acoustic Spectroscopy (QEPAS). The variability of the environment (temperature, humidity) can lead to a drift of the QTF resonance. In most applications, regular QTF calibration is absolutely essential. Because the requirements vary greatly depending on the field of application, different characterization methods can be found in the literature. We present a review of these methods and compare them in terms of accuracy. Then, we further detail one technique, called Beat Frequency analysis, based on the transient response followed by heterodyning. This method proved to be fast and accurate. Further, we demonstrate the resonance tracking of the QTF while changing the temperature and the humidity. Finally, we integrate this characterization method in our Resonance Tracking (RT) QEPAS sensor and show the significant reduction of the signal drift compared to a conventional QEPAS sensor.

## 1. Introduction

The quartz tuning fork (QTF) is a piezoelectric transducer with a high quality factor (above 10,000 at atmospheric pressure). It was originally manufactured by the watch industry as a standard 32,768 Hz clock oscillator. Industry standard made the QTF reliable, available and affordable. Owing to its great features, it was employed as the sensitive element in many fields and systems:Atomic Force Microscopy (AFM), near-field, microwave microscopy [1,2,3,4]mass/viscosity sensor [5,6].gas sensing using Quartz-Enhanced Photo-Acoustic Spectroscopy (QEPAS) [7]

The QTF resonance is related to the QTF intrinsic properties, as well as the environmental conditions. Any modification in those parameters affect the QTF behavior. The intrinsic properties are physical features (geometry, density, Young’s modulus, …) and present long-term stability, whereas the environmental conditions (temperature, pressure, humidity, gas density, …) are prone to quick variations. The effect of these on the resonance frequency *f*_0_ and the quality factor Q of the QTF, can be individually quantified.

Depending on the application and the selected electronic circuit, the QTF can be operated in different modes, leading to different mesurands: QTF current amplitude, frequency shift, and Q shift, etc. The QTF output current is typically about a few fA to a few μA, amplification stages are thus needed prior to readout. Various approaches are known to follow the QTF resonance, depending on the application and using different electronics circuits. Details of the characterization methods and a literature review of their performances are presented in Section 2.

QEPAS is a gas sensing technique that was proposed in 2002 by Kosterev et al. [7]. As in photoacoustics, a modulated laser is used to excite a gas specie in order to create a sound wave thanks to the photoacoustic effect. Then, the acoustic energy is converted to an electrical signal by the piezoelectric QTF. The QEPAS technique was successfully employed for detecting various gas species [8]. Some alternative techniques were developed to improve the sensitivity such as the addition of micro-tubes [9], the use of a laser cavity [10] or making custom QTFs [11]. However, the reliability of the QEPAS sensor is barely discussed in the literature. A key point for robustness is the sensitivity of the QTF parameters to the environment. A method is thus required to measure the QTF parameters, *f*_0_ and Q, with sufficient accuracy and short timing.

For an optimum sensitivity, the QTF must be operated at its resonant frequency. For our QEPAS application, the main purpose is to obtain a sensor that has high sensitivity to the target gas and is independent of the environmental conditions. Based on the previous study in Section 2, we selected the Beat Frequency (BF) technique for tracking the resonance of the QTF. First, the principle of operation of the BF technique is described (Section 3). Then, the resonance tracking is tested by observing the effect of temperature and humidity (Section 4). Finally, we compare the response, to a humid gas injection, of our Resonance Tracking (RT) QEPAS sensor with the conventional QEPAS (Section 5).

## 2. QTF Characterization Methods

In AFM microscopy and viscosity sensing, a standard approach is the oscillator circuit. Active components like transistors and operational amplifiers are used as amplifiers and the QTF acts as a band pass filter. For the circuit to start oscillating, particular conditions, called Barkhausen conditions, must be fulfilled. This implies an adjustment of the phase and gain of the feedback loop, leading to more complex circuits [12]. This feedback loop also affects the QTF behavior. For instance, the quality factor can be artificially increased [13].

In photoacoustics, the electronics circuit is generally very basic. It is based on a transimpedance amplifier which converts the QTF piezoelectric current into a readable voltage. A low-noise and high bandwidth operational amplifier and an appropriate printed circuit board design are the key points of the electronics. Distinguishably from the first method, the QTF is normally at rest, no external energy is provided. Also, the QTF output current is monitored as the output signal, instead of the frequency shift. This simple and easy-to-use configuration has a major drawback: no feedback ensures a resonant operation.

In laboratory, the environmental conditions can be stabilized in order to have the QTF resonance stable during a complete experiment. However, in industrial and outdoor environments, the resonance has to be checked regularly. The period of verification depends on the speed of change of the parameters.

There are different techniques for the measurement of *f*_0_ and Q:Frequency sweep (Figure 1a): the QTF is excited by applying a frequency-swept sinusoid with a function generator, and the output amplitude is measured with an oscilloscope after a trans impedance amplification stage. The scan speed is limited by the QTF response time, a few hundred milliseconds. The overall measuring time is generally more than 5 s. Alternatively, it can be done with an impedance analyzer, but it implies additional cost and electronics. *f*_0_ and Q are obtained through the fitting of the response with the usual Butterworth-Van Dyke model [14].Transient analysis or “Ring-down” (Figure 1b): It consists of two steps: excitation and relaxation. During the excitation, energy is provided to the QTF using a sinus, a pulse or a broadband white noise signal. Then the excitation source is turned off. The QTF instantly oscillates at his fundamental frequency, and slowly disperses the stored energy with a typical exponential decay whose decay amplitude is related to the quality factor. It has already been employed in photoacoustics [15].Beat Frequency (BF) analysis (Figure 1c): It stems from the transient analysis with an additional step of demodulation. The high-frequency signal is translated to a low-frequency sine wave. With the latter, *f*_0_ can be measured with great accuracy. When demodulation is made through a lock-in amplifier, the time constant must be short enough to avoid signal distortion, that would lead to erroneous values.Oscillator circuit: extensive theoretical models, electronics circuit and operating principles can be found in the literature, for instance in [3,16]. The measurement of *f*_0_ is fast and accurate. However, the quality factor cannot be measured and is often calculated beforehand, for calibration purposes, using the frequency sweep method.

Frequency sweep is often preferred because it requires usual benchtops instruments. However, it is too slow for real time measurements, caused by the long QTF time response. Also, additional capacitances can induce deviations in the response from the theoretical one, leading to poor accuracy of the deduced parameters. To remedy it, a compensation circuit must be added [17]. Transient and BF analysis rely on the same physical principle, only the signal treatment differs. In both cases, the Q value is obtained from the signal envelope with a relatively good accuracy. The output of the transient analysis is at 32.7 kHz and requires high speed digitizing in order to obtain an accurate value for *f*_0_. Heterodyning consists in mixing the unknown signal with a reference signal of a known frequency. The output is a low frequency beat signal. The uncertainty on *f*_0_ is drastically reduced. It mostly improves the accuracy on the measurement of *f*_0_. It is worth mentioning that, even with the same setup, discrepancies in the results are observed between the different methods [6,18].

The most relevant results about the measurement error made on *f*_0_ and Q, for a given measurement time, are summarized in Table 1. In mass/viscosity sensing, the main concern is the monitoring of the resonant frequency; as values for Q are often inexistent. Heterodyning is a well-known and high-accuracy technique in frequency stability analysis [20] and can be a very suitable technique for measuring the QTF parameters. For our application, an accurate value for both *f*_0_ and Q was needed, with a measurement time inferior to one second. After the comparison of the methods, the BF analysis was judged to be the most appropriate. This BF technique was originally experimented using a photoacoustic pulsed excitation [21]. As a consequence, the excitation amplitude depends on the analyte concentration. This means at low concentration the output signal is noisy, and the measurement of the QTF parameters can be very inaccurate. That is why, we preferred to use an electrical instead of photoacoustic excitation.

## 3. Principle of Operation

Beat Frequency was selected for measuring the QTF parameters, and here we describe here its principle of operation, first expressing the transient response, and then deducing the BF signal. The purpose of the measurement is to be able to quantify the value of *f*_0_ and Q, with sufficient accuracy and timing. The total duration of the measurement should be less than a 1 s, desirable dead time for our gas sensor. The required accuracy is 0.2 Hz and 40, resp. for *f*_0_ and Q. This corresponds to a 1% deviation in the gas concentration signal, considering the QTF response as a lorentzian curve centered at *f*_0_ (32.7 kHz) and with a full width at half maximum of *f*_0_/Q (Q = 8000). The BF technique was thus chosen to meet these requirements of accuracy. It is based on a two-step procedure: excitation and relaxation (Figure 2). The first step is to drive the QTF in a forced oscillation regime, at a fixed frequency *f*_exc_, whereby energy is provided to the resonator. The closer *f*_exc_ is to *f*_0_, the more energy will be accumulated. However, *f*_0_ is still an unknown parameter. Accordingly, *f*_exc_ was chosen to be a few Hz off the estimated moving span of *f*_0_ during the system operation.

Once the QTF has reached a steady state at *t* = *t*_exc_, that means at least 3τ (τ the time constant of the QTF), the excitation source is switched off and the relaxation takes place. The transient response can be found with the differential equation of a harmonic oscillator and the solution is a simple damped oscillation (Figure 2b):(1)$$V_{out}\left(t\right)={\hat{V}}_{out}\sin\left[2\pi f_{0}\left(t-t_{exc}\right)\right]exp\left[-\frac{\pi f_{0}}{Q}\left(t-t_{exc}\right)\right],\;\left(t>t_{exc}\right)$$where ${\hat{V}}_{out}$ is the steady-state output voltage amplitude after excitation. The information of *f*_0_ and Q are embedded, respectively, in the carrier and the envelope of the relaxation signal.

Different approaches can be used in order to extract the information from the signal. It should be underlined that the most critical part is about achieving a 0.2 Hz accuracy for a 32 kHz sinus. Using a counter clocked at a few MHz could be a solution, but would not give any result for Q. A purely numerical solution, based on high-speed digitizing followed by post processing with a FPGA would probably give the most compact system but requires programming expertise and time resources.

In our case, the signal was extracted in a heterodyne configuration using a lock in amplifier (LIA). It is convenient because the LIA is also employed for the detection of the photoacoustic signal by harmonic detection, as common practice to increase the signal-to-noise ratio. The LIA realizes the demodulation of the output signal by the reference signal of frequency *f*_ref_ giving rise to a beating at a frequency *f*_0_–*f*_ref_. In order to avoid the LIA low-pass filtering, the bandwidth is selected to be broader than the beat frequency (BF), thus the time constant is typically taken to be around 1 ms [21].

After demodulation, the BF signal can be simply written (Figure 2c):(2)$$V_{BF}\left(t\right)={\hat{V}}_{out}\sin\left[2\pi |f_{0}-f_{ref}|\left(t-t_{exc}\right)\right]exp\left[-\frac{\pi f_{0}}{Q}\left(t-t_{exc}\right)\right],\;\left(t>t_{exc}\right)$$

The BF signal is a low-frequency signal which can easily be digitized and fitted by the Equation (2) in order to obtain the QTF parameters.

For our first experiments, we have taken *f*_ref_ equal to *f*_exc_. But since those two frequencies do not play the same role in the BF they could be conveniently optimized at different values. The excitation frequency could be selected to maximize the excitation of the QTF, thus having a value quite close to the resonant frequency. The reference frequency could be set to get a beat frequency from which the resonant frequency (*f*_0_) could be extracted with good accuracy and that would minimize the LIA bandwidth to reduce the noise.

## 4. Effect of Temperature and Humidity

### 4.1. Experimental Setup

The scheme of the setup is presented on Figure 3. The driving signal for the excitation was provided by a function generator (Tektronix AFG 1062), at a given frequency *f*_exc_. A 12 V DPDT relay (Hongfa HDF31, Radiospares) was used to switch between the excitation source and the ground. The relay was controlled by a data acquisition card (DAQ) (Labjack T7, Lakewood, CA, USA). The QTF, a common NC32LF, oscillating around 32,756 Hz, was followed by a homemade transimpedance amplifier with a 10 MΩ feedback resistor and a lock-in amplifier (EG&G 7260). Shielded cables were necessary in order to prevent electromagnetic radiation during QTF relaxation, that would have otherwise caused a large signal background. The QTF frequency was first estimated by a quick frequency scan. The excitation frequency was then set 20 Hz below this value. The excitation voltage *V*_exc_ was set to obtain a signal amplitude of about 1 volt after the transimpedance, ensuring a high signal-to-noise ratio and thus improving the accuracy of the measurement.

In order to get the BF signal, the time constant and the filter slope of the lock-in amplifier were respectively 640 μs and 12 dB/oct. A Labview (National Instruments) program collected the data from the LIA, directly followed by a signal treatment algorithm, thus obtaining *f*_0_ and Q.

The algorithm had to fulfill the criteria we had set before: 1% signal error and 1 s measurement time. For our labview program, we decided to avoid using a nonlinear fitting, but rather a combination of a numerical low pass filtering and peak detection. It proved to be faster and as accurate. For testing the algorithm, a function generator was used to generate an exponentially decaying sine wave, similar to the QTF relaxation profile. The accuracy was measured to be below 0.01 Hz and 10, resp for *f*_0_ and Q, in ideal conditions, but reached 0.1 Hz and 100 in practice with the QTF. This discrepancy was mostly due to the signal noise and the relay switching disturbance. With our approach, the criterion was absolutely met for *f*_0_, and almost for Q. Future work will focus on improving the performance for Q value assumption.

The effect of temperature and humidity were characterized in 2 different experiments:Exp 1: The QTF temperature behavior was observed using a homemade temperature regulated system. A commercial PID temperature controller was driving a thermoelectric cooler at a given temperature. The QTF metal cap was thermally connected to the cold plate of the thermoelectric cooler. In this way, the air gap between the QTF and the temperature-controlled cap was small, ensuring high thermal conductivity, and thus fast response time. Originally manufactured under vacuum, the QTF cap was punctured to obtain atmospheric pressure when needed.Exp 2: The QTF humidity behavior was assessed with a benchtop humidity chamber (ESPEC SH-242) It was operated at a constant temperature of 20 °C. The humidity could be varied from 30 to 90% Relative Humidity (RH). The chamber response was in the range of a few minutes, and thus each humidity step was hold for 30 min.

### 4.2. Results

First, the QTF behavior is studied as a function of temperature (Exp 1). A temperature ramp, ranging from 10 to 60 °C, with 2.5 °C steps, was applied to the QTF, while recording the resonant frequency. Two different QTFs were tested, one at atmospheric pressure and one under vacuum capped. As described previously, the setup was engineered in order to ensure good thermal conductivity. The frequency response to a temperature step is displayed as inset in Figure 4a and exhibits a fast response (about 20 s) as expected. The average value of every step was calculated and plotted as a function of the temperature (Figure 4a). The curves show a smooth and almost perfect parabolic response as predicted by theoretical models [22,23]. This corresponds very well to the description in the QTF specification sheet. The parabola is centered at T = 22.5 °C (*f*_0_ = 32,764.20 Hz) for the vacuum QTF and at T = 30 °C (*f*_0_ = 32,757.75 Hz) for the open QTF. The frequency shift, as the pressure increases, can be explained by the increased density and viscosity of the surrounding fluid [24].

Exp 1 gave us nice results that could be compared to manufacturing specifications, and was thus an intermediate experiment in order to validate the accuracy and reliability of our QTF characterization setup.

In Exp 2, the humidity was varied from 30% to 90% RH with a 10% RH step, followed by two steps at 50% and 30% RH. *f*_0_ shows a shift of about 2 Hz from 30% to 80% RH, and almost 1 Hz from 80 to 90% RH. Then, the signal well recovered a higher *f*_0_ for the decreasing humidity steps. It can be noted as well that the temperature was constant throughout the experiment except a small increase for the 90% RH step, which should lead to a minor shift compared to the humidity-related shift. The quality factor is also affected by the humidity. It decreases as the humidity increases, with linear behavior whereas the resonance frequency has a quadratic one. Q shifts of about 300 over the whole range of humidity.

Those results enhance the importance to take into account the temperature and humidity in a sensor employing a QTF as a sensitive transducer. We have proposed a setup based on BF QEPAS to quantify the variations of these parameters. The curves can be used to determine the gas sensor error for a given temperature and humidity range. They can also be used as calibrated curves in order to make a temperature/humidity compensated sensors, and thus to improve the sensor performances. Another option is to employ the BF technique for real time resonance tracking as it is presented in the next paragraph.

## 5. QEPAS with QTF Resonance Tracking

After having successfully implemented the BF analysis for measuring the QTF parameters and its variations with temperature and humidity, the next step was to add it to the gas sensor to continuously monitor the changes. A post-processing approach would be to use the results of the BF measurement to normalize the output signal. Another approach would be to add a feedback loop to ensure the resonant operation of the QTF and thus maximizing the output signal. Here, we present this last method, we called it Resonance Tracking (RT) QEPAS.

The gas sensor is based on the QEPAS technique, with a DFB laser (NORCADA), for the detection of methane at 4294.55 cm^−1^ (2328.53 nm). The laser is operated at 20 °C and 143.4 mA. The current is wavelength-modulated with a sine wave at 32.7 kHz and covering 0.8 cm^−1^. For gas detection, the QTF is connected to the transimpedance amplifier on one side and grounded to the other side (for QEPAS) or connected to the function generator (for BF characterization).

The goal is to sequentially measure the gas concentration and the QTF parameters. First, the QTF parameters are obtained using the BF analysis as described in the section “Principle of operation”. Then, the laser modulation frequency is modified to match with the QTF instantaneous frequency. Finally, the gas concentration is measured.

There are a few considerations for implementing properly RT-QEPAS. The electrical excitation is greater than the one imposed by acoustic force by a few orders of magnitude. It implies in practice that it is not necessary to turn off the laser during the characterization of the QTF. This continuous current operation favors the laser emission stability, so it is profitable for QEPAS. The synchronization of all the instruments is crucial for RT-QEPAS, it was done through a Labview program with a state machine architecture. The program also conveniently allowed the user to quickly switch between the QTF characterization and gas sensing, enabling fast troubleshooting. During the BF measurement, the LIA constant and the reference signal were set to 1 ms and to 32,731 Hz whereas during QEPAS the were changed to 100 ms and 32,751 Hz (or measured *f*_0_ for RT QEPAS). The duration of the overall measurements was mostly limited by the QTF relaxation time after the electrical excitation. Due to the high Q value and the large excitation voltage, it takes not less than one second for the QTF to go under an energy threshold that could be considered low enough compared to the acoustic energy. It can be easily computed by taking the envelope of equation (2) and calculating the decay time of the QTF between the voltage at *t*_exc_ and the threshold voltage at *t*_relax_. For this first implementation of RT-QEPAS, the total cycle time took 3.2 s including the QTF characterization (0.2 s excitation + 1 s relaxation), QEPAS measurement (1 s), and post processing (1 s). It can be drastically improved by separating the post-processing from the measurement and fine-tuning the code.

The humidity affects the sensor through two main mechanisms: modification of the relaxation time and modification of the QTF resonance. We have seen in Section 4 the impact of humidity on the QTF parameters. Due to the long relaxation time of methane, the photoacoustic effect in the 30 kHz frequency range is not very effective. H_2_O promotes non-radiative relaxation and thus increase the efficiency of the photoacoustic effect. It is thus relevant trying to increase the water content of the sample. A simple way of humidifying a gas is to use a bubbling bath. The gas passes through the water and the humidity increases. The humidity level depends essentially on the flux and the bath temperature. The water bath has a beneficial regulatory effect in systems prone to quick variations, for instance in a breath analysis sensor.

In this experiment, we observe the response of the sensor to a gas cycle including a step of 1% dry CH_4_ followed by a step of 1% wet CH_4_. The gas cycle is repeated twice, first with standard QEPAS, and second with RT-QEPAS (Figure 5). The QTF parameters are also recorded. In both cases, the injection of dry CH_4_ leads to the same QEPAS signal. However, during the wet CH_4_ step, the QEPAS signal (green curve) exhibits a strong decrease whereas the RT-QEPAS one (blue curve) remains practically constant. The QEPAS signal drift creates a measurement artifact, which can be chiefly explained by the frequency shift of the QTF (Figure 4). The shift of about 3 Hz implies in the case of QEPAS, a 44% relative change in amplitude. In RT-QEPAS, a 2% variation of the signal is observed. Then the signal can be normalized by Q (red curve), leading to less than a 1% signal shift, which is a significant reduction of the error due to the frequency shift of the QTF. This normalization further improves the shape of the signal and corrects the overshoot of the RT-QEPAS signal (as shown in the inset). The normalized signal response follows an exponential growth. To conclude with, this experiment shows the whole potential of the resonance tracking method to correct the QEPAS signal, using both *f*_0_ and Q, and thus limiting the signal drift due to the non-controllable variations of environmental parameters.

## 6. Conclusions

The QTF is a key element in a QEPAS gas sensor. We analyzed the different techniques for the characterization of the QTF resonance and presented more thoroughly the Beat Frequency measurement with electrical excitation. This technique has considerable advantages, including the simultaneous measurement of *f*_0_ and Q, constant excitation amplitude and use of conventional benchtop instruments. The performances of the method were verified by observing the influence of the temperature and humidity. The obtained responses can be used as calibration curves. In our cases, the QTF characterization was implemented in a feedback loop in order to ensure the resonance tracking of the QTF. RT-QEPAS showed a consistent reduction of the signal drift, less than 1% relative error compared to 44% for conventional QEPAS. Future work will focus on the reduction of the BF measurement time, chiefly through the optimization of the algorithm and the Labview structure. Another interesting idea is to implement an electronic damping circuit that would fasten the QTF relaxation and thus reduce the overall measurement time.

## Figures and Tables

**Figure 1 sensors-19-05565-f001:**
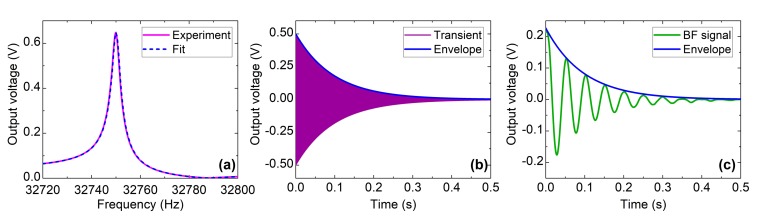
The same QTF is characterized with a frequency sweep (*f*_0_ = 32,750.0 Hz, Q = 9783) (**a**), the transient analysis (Q = 10,031) (**b**) and the BF analysis (*f*_0_ = 32,750.3 Hz, Q = 9263) (**c**). The experimental is fitted with the electrical model for (**a**) and with an exponentially decaying sinusoid for (**c**). The signal envelop is extracted in (**b**) in order to get a value of Q.

**Figure 2 sensors-19-05565-f002:**
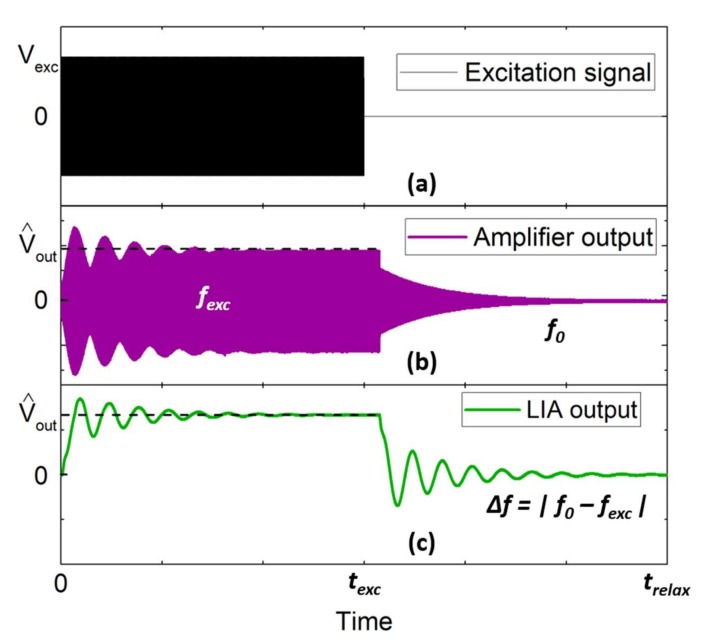
Observation of typical signals in BF analysis: the excitation signal (**a**), the output of the transimpedance amplifier (**b**) and the output of the lock-in amplifier (**c**). The excitation signal provides the initial energy to the QTF with a sinusoid at a frequency *f*_exc_ during a time *t*_exc_. The QTF is forced to oscillate at *f*_exc_, leading to a continuous demodulated signal on the lock-in amplifier (LIA) output. When the excitation signal drops to zero, the QTF returns to its natural frequency *f*_0_, resulting in a Beat Frequency (BF) signal on the LIA.

**Figure 3 sensors-19-05565-f003:**
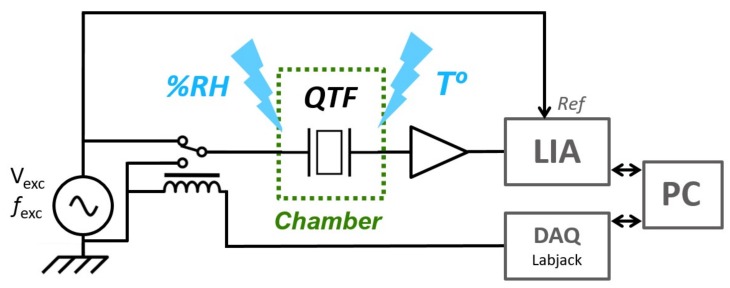
Setup for the BF measurement. The QTF is enclosed in a temperature and humidity regulated chamber. The relay, controlled by the analog output of a DAQ card, is used to switch between the excitation source and the ground. The QTF current is amplified and then demodulated by a lock-in amplifier. A labview program on a laptop synchronizes the instruments to obtain the BF signal, from which are extracted the QTF parameters in real time.

**Figure 4 sensors-19-05565-f004:**
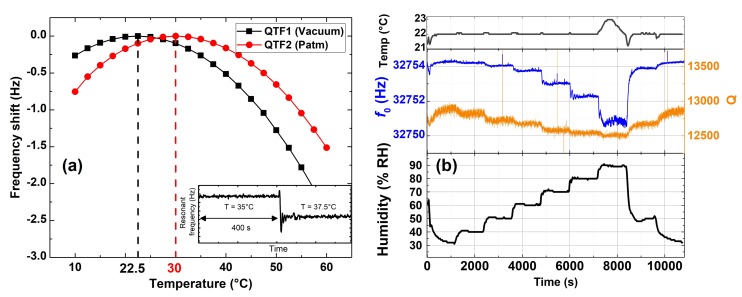
(**a**) Resonant frequency of a vacuum capped (black) and an open QTF (red) as a function of temperature. The inset (bottom right) represents the resonant frequency versus the time for two temperature steps. (**b**) Recording the QTF parameters *f*_0_ and Q while varying the humidity and keeping the temperature constant. The humidity cycle is made of 10% RH steps: 7 ascending steps (30 to 90% RH) and 2 descending steps (50% and 30% RH).

**Figure 5 sensors-19-05565-f005:**
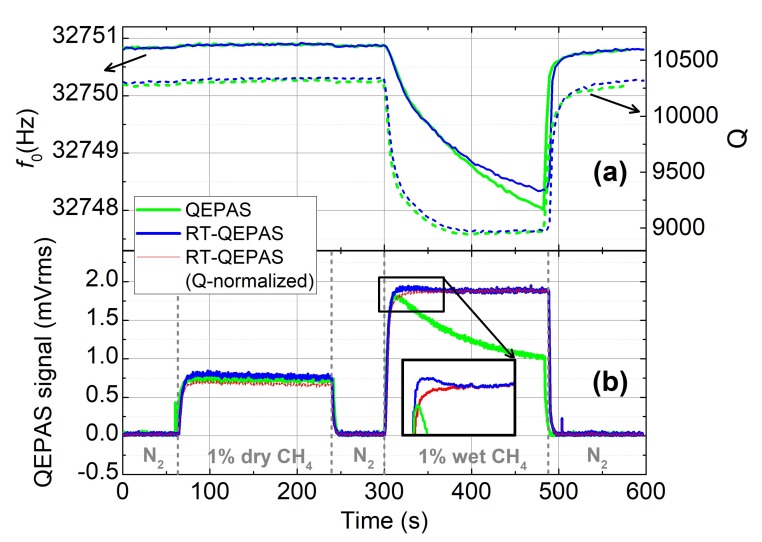
(**a**) Evolution of the resonant frequency (continuous line) and the quality factor (dotted line) and (**b**) response of the gas sensor to an injection of 1% dry CH_4_ and 1% wet CH_4_, with the QEPAS (green) and the RT-QEPAS technique. In RT-QEPAS, the QTF instantaneous frequency *f*_0_ is used as a feedback for the laser modulation frequency (blue), and then normalized by the Q factor (red). The RT-QEPAS sensor is thus more robust to environment changes than the conventional QEPAS sensor. The gas cell is flushed with pure nitrogen between the two injections.

**Table 1 sensors-19-05565-t001:** Summary of the measurement accuracy on *f*_0_ and Q with different characterization techniques.

Characterization Method	*f*_0_ Error (Hz)	Q Error	Measurement Time (s)	Reference
Frequency response (from FFT)	0.065	470	10	[17]
Frequency response	6.6	/	/	[6]
Transient analysis	26	590	/	
Oscillator circuit	0.001 (Theoretical)	/	0.1	[19]
Oscillator circuit	0.01	/	1	[5]
	0.001	/	10	

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
