# Peer review of "Quartz Tuning Fork Resonance Tracking and application in Quartz Enhanced Photoacoustics Spectroscopy"

_sensors, 2019, doi:10.3390/s19245565_

Round 1
Reviewer 1 Report
In this paper, the authors report on the application of a technique called Beat Frequency analysis, based on the transient response followed by heterodyning to track the quartz tuning fork (QTF) resonance frequency. The propose technique has been demonstrated in changing temperature and humidity, and finally applied to QEPAS signal for CH4 gas concentration measurements. The paper is written and organized well. I have very few complaints with this manuscript. However, I would like the authors to consider a few points when polishing up the final draft.
The quartz tuning fork (QTF) has been a versatile component for developing various transducers in many fields, it was employed as the sensitive element in many fields:Atomic Force Microscopy (AFM)/Near-field microscopy;mass/viscosity sensor; Quartz-Enhanced Photo-Acoustic Spectroscopy (QEPAS).
I was a little disappointed with the lack of appropriate references in the introduction. There has been many excellent work done in other fields, such as QTF based photoelectric detector for absorption spectrosocpy and standoff spectroscopy, at least two stand-out papers might be included in my opinion:Optics and Lasers in Engineering, 115:141-148,2019; Review of Scientific Instruments, 87: 123101-6, 2016.
The authors state that "the humidity affects the sensor through two main mechanisms: modification of the relaxation time and modification of the QTF resonance". Indeed, H2O has already been found to be an effective foreign broadener for CO2 with near 1.57 μm [J. Phys. Chem. A 115 (2011) 13804-13810] and typically twice greater than those by air at 4.3 μm [Can. J. Phys. 87(2009) 469-484.], 3-4 times higher for six ammonia (NH3) transitions lines near 1103.46 cm-1 [J. Quant. Spectrosc. Radiat. Transfer 121 (2013) 56-68], 1.8 and 1.9 times higher for CO and N2O detection near 4.57 μm [Journal of Molecular Spectroscopy, 331:34-43, 2017]. Moreover, the published results show that H2O-broadenings of methane (CH4) lines are, on average, 34% larger than those for dry air [J. Quant. Spectrosc. Radiat. Transfer 173 (2016) 40-48]. Therefore, the diluting effect and broadening effect cannot be neglected for high-accuracy gas concentration retrievals. In this work, the humidity was varied from 30 to 90% RH. How to resolve this issue ? and this effect shoud also be discussed in the paper.
Reviewer 2 Report
The manuscript “Quartz Tuning Fork Resonance Tracking and application in Quartz Enhanced Photoacoustics Spectroscopy” by Roman Rousseau et al. presented a review of different QTF calibration methods and compare them in terms of accuracy. The manuscript also described a resonance tracking QEPAS sensor based on beat frequency QEPAS technique. The article is clearly written and provides convincing experimental results. From my point of view, I feel it can be published in Sensors after the authors finish some necessary modifications.
In line 121, the authors stated that “there were, to our knowledge, no study on the performances of the BF technique for measuring the QTF parameters”. It’s WRONG! As mentioned in Ref. 19, Wu et al. has studied beat frequency technique in detail and stated that this technique can be used to calibrate the frequency and the quality factor of the QTF. The contribution of this manuscript is using the beat frequency technique for measuring the QTF parameters and its variations with temperature and humidity. Please correct it. In line 131, the authors stated that “first expressing the transient response”. It’s wrong. There are many articles in which the transient response of the QTF has been studied, for example, Ref. 19. Please check the whole article to correct the relevant descriptions. The paper “Mordmuller. M., Kohring. M., Schade. W. and Willer. U. An electrically and optically cooperated QEPAS device for highly integrated gas sensors. Appl. Phys. B 119, 2015” should be cited as its authors studied the method of the QTF electrical excitation for the first time. In line 134, the authors stated that “The required accuracy is 0.2Hz and 40, resp. for f0 and Q. This corresponds to a 1% deviation in the gas concentration signal, considering the QTF response as a Lorentzian curve centered at f0 (32.7kHz) and with a full width at half maximum of f0/Q (Q=8000).” Please add some experiments or figure to verify it. The concentration of the target gas was 10000 ppm. When you check the performance of a gas sensor, you should use a low concentration sample. What’s the detection sensitivity of this RT-QEPAS? There are some obvious spelling errors: in line 21, avoid the first comma, it is not necessary; line 292, the “CH4” should be “CH4”. Please check the whole article. Please correct the information about the Ref. 19. The order of the authors is totally wrong.Author Response
Please see the attachment.

Reviewer 3 Report
The authors reviewed the QTF technology with accuracy comparison called Beat Frequency analysis, based on the transient response followed by heterodyning. Before publication, the authors should address following issues.
1. Check the english issues with care.
2. Better to mention what is the main advantage of usage of BF technique for QTF characterization.
3. Better to mention about the reason that maximum value of frequency shift (0Hz) is located in 22.5 degC for vacuum, 30 degC for ambient in Fig. 4a.
4. When the humidity goes above 70%, the QTF signal (freq. shift) shows unstable behavior in Fig. 4b., better to mention some clues about this phenomenon.
5. Can you explain clearly that the signal drifting of QEPAS for wet CH4 1% applying is just artifact? Is it real nature of chamber condition when wet CH4 1% is applied? Is it always be an issue of the QEPAS measurement before?
6. Better to show another gas materials for testing performance of the proposed device (not mandatory).
Reviewer 4 Report
The manuscript is of good level Temperature graph in Fig.4 (b) is clipped at 22.2°C, but the temperature value it’s above that value. The graph should be corrected.Author Response
The authors re scaled the temperature graph appropriately in Figure 4.b.
Thank you for the review.
Round 2
Reviewer 2 Report
The authors have revised the manuscript carefully. And this paper can be accepted after two relevant paper be referenced.
The paper "Atmospheric CH4 measurement near a landfill using an ICL-based QEPAS sensor with V-T relaxation self-calibration" (Sensors & Actuators: B. Chemical,2019, 297,126753) and "Quartz enhanced photoacoustic H2S gas sensor based on a fiber-amplifier source and a custom tuning fork with large prong spacing" (Applied Physics Letters,2015, 107, 111104) should also be referenced as Ref. 12 and Ref. 13 in line 393.
Author Response
The papers were added as Ref 12 and 13, as suggested by the reviewer.